# Manifold Forests: Closing the Gap on Neural Networks

## Abstract

Decision forests (DF), in particular random forests and gradient boosting trees, have demonstrated state-of-the-art accuracy compared to other methods in many supervised learning scenarios. In particular, DFs dominate other methods in tabular data, that is, when the feature space is unstructured, so that the signal is invariant to permuting feature indices. However, in structured data lying on a manifold—such as images, text, and speech—neural nets (NN) tend to outperform DFs. We conjecture that at least part of the reason for this is that the input to NN is not simply the feature magnitudes, but also their indices (for example, the convolution operation uses "feature locality"). In contrast, naïve DF implementations fail to explicitly consider feature indices. A recently proposed DF approach demonstrates that DFs, for each node, implicitly sample a random matrix from some specific distribution. Here, we build on that to show that one can choose distributions in a *manifold aware fashion*. For example, for image classification, rather than randomly selecting pixels, one can randomly select contiguous patches. We demonstrate the empirical performance of data living on three different manifolds: images, time-series, and a torus. In all three cases, our Manifold Forest (MORF) algorithm empirically dominates other state-of-the-art approaches that ignore feature space structure, achieving a lower classification error on all sample sizes. This dominance extends to the MNIST data set as well. Moreover, both training and test time is significantly faster for manifold forests as compared to deep nets. This approach, therefore, has promise to enable DFs and other machine learning methods to close the gap with deep nets on manifold-valued data.

## 1 Introduction

Decision forests, including random forests and gradient boosting trees, have solidified themselves in the past couple decades as a powerful ensemble learning method in supervised settings (Fernández-Delgado et al., 2014; Caruana & Niculescu-Mizil, 2006), including both classification and regression (Hastie et al., 2001). In classification, each forest is a collection of decision trees whose individual classifications of a data point are aggregated together using majority vote. One of the strengths of this approach is that each decision tree need only perform better than chance for the forest to be a strong learner, given a few assumptions (Schapire, 1990; Biau et al., 2008). Additionally, decision trees are relatively interpretable because they can provide an understanding of which features are most important for correct classification (Breiman, 2001). Breiman originally proposed decision trees that partition the data set using hyperplanes aligned to feature axes (Breiman, 2001). Yet, this limits the flexibility of the forest and requires deep trees to classify some data sets, leading to overfitting. He also suggested that algorithms which partition based on sparse linear combinations of the coordinate axes can improve performance (Breiman, 2001). More recently, Sparse Projection Oblique Randomer Forest (SPORF), partitions a random projection of the data and has shown impressive improvement over other methods (Tomita et al., 2015).

Yet random forests and other machine learning algorithms frequently operate in a tabular setting, viewing an observation $\vec{x} = (x_1, \ldots, x_p)^\mathsf{T} \in \mathbb{R}^p$ as an unstructured feature vector. In doing so, they neglect the indices in settings where the indices encode additional information. For structured data, e.g. images or time series, traditional decision forests are not able to incorporate known continuity between features to learn new features.

For decision forests to utilize known local structure in data, new features encoding this information must be manually constructed. Prior research has extended random forests to a variety of computer vision tasks (Lepetit et al., 2005; Gall et al., 2011; Bosch et al., 2007; Shotton et al., 2011) and augmented random forests with structured pixel label information (Kontschieder et al., 2011). Yet these methods either generate features a priori from individual pixels, and

thus do not take advantage of the local topology, or lack the flexibility to *learn* relevant patches. Decision forests have been used to learn distance metrics on unknown manifolds (Criminisi et al., 2012), but such manifold forest algorithms are unsupervised and aim to learn a low dimensional representation of the data.

Inspired by SPORF, we propose a projection distribution that takes into account continuity between neighboring features while incorporating enough randomness to learn relevant projections. At each node in the decision tree, sets of random spatially contiguous features are randomly selected using knowledge of the underlying manifold. Summing the intensities of the sampled features yields a set of projections which can then be evaluated to partition the observations. We describe this proposed classification algorithm, Manifold Forests (MORF) in detail and show its effectiveness in three simulation settings as compared to common classification algorithms. Furthermore, the optimized and parallelizable open source implementation of MORF in R and Python is available. This addition makes for an effective and flexible learner across a wide range of manifold structures.

## 2 BACKGROUND AND RELATED WORK

### 2.1 CLASSIFICATION

In the two-class classification setting, there is a data set $D = \{(x_i, y_i)\}_{i=1}^n$ of n pairs $(x_i, y_i)$ drawn from an unknown distribution $F_{XY}$ where $x_i \in \mathcal{X} \subset \mathbb{R}^p$ and $y_i \in \mathcal{Y} = \{0, 1\}$. Our goal is to train a classifier $h(x; D_n) : \mathcal{X} \times (\mathcal{X} \times \mathcal{Y})^n \to \mathcal{Y}$ based on our observations that generalizes to correctly predict the class of an observed $x$. The performance of this classifier is evaluated via the 0-1 Loss function $L(h(x), y) = \mathbb{I}[h(x) \neq y]$ to find the optimal classifier $h^* = \operatorname{argmin}_h \mathbb{E}[L(h(x), y)]$, which minimizes the probability of an incorrect classification.

### 2.2 RANDOM FORESTS

Originally popularized by Breiman, the random forest (RF) classifier is empirically very effective (Fernández-Delgado et al., 2014) while maintaining strong theoretical guarantees (Breiman, 2001). A random forest is an ensemble of decision trees whose individual classifications of a data point are aggregated together using majority vote. Each decision tree consists of split nodes and leaf nodes. A split node is associated with a subset of the data $S = \{(x_i, y_i)\} \subseteq D$ and splits into two child nodes, each associated with a binary partition of $S$. Let $e_j \in \mathbb{R}^p$ denote a unit vector in the standard basis (that is, a vector with a single one and the rest of the entries are zero) and $\tau$ a threshold value. Then $S$ is partitioned into two subsets given the pair $\theta_j = \{e_j, \tau\}$.

$$S_\theta^L = \{(x_i, y_i) \mid e_j^\mathsf{T} x_i < \tau\}$$
$$S_\theta^R = \{(x_i, y_i) \mid e_j^\mathsf{T} x_i \geq \tau\}$$

To choose the partition, the optimal $\theta^* = (e_j^*, \tau^*)$ pair is selected via a greedy search from among a set of $d$ randomly selected standard basis vectors $e_j$. The selected partition is that which maximizes some measure of information gain. A typical measure is a decrease in impurity, calculated by the Gini impurity score $I(S)$, of the resulting partitions (Hastie et al., 2001). Let $\hat{p}_k = \frac{1}{|S|} \sum_{y_i \in S} \mathbb{I}[y_i = k]$ be the fraction of elements of class $k$ in partition $S$, then the optimal split is found as

$$I(S) = \sum_{k=1}^K \hat{p}_k (1 - \hat{p}_k)$$
$$\theta^* = \operatorname*{argmax}_\theta |S| I(S) - |S_\theta^L| I(S_\theta^L) - |S_\theta^R| I(S_\theta^R).$$

A leaf node is created once the partition reaches a stopping criterion, typically either falling below an impurity score threshold or a minimum number of observations (Hastie et al., 2001). The leaf nodes of the tree form a disjoint partition of the feature space in which each partition of observations $S_k$ is assigned a class label $c_k^*$ corresponding to the class majority.

$$c_k^* = \operatorname*{argmax}_{c_k} \sum_{y_i \in S} \mathbb{I}(y_i = c_k).$$

A decision tree classifies a new observation by assigning it the class of the partition into which the observation falls. The forest averages the classifications over all decision trees to make the final classification (Hastie et al., 2001). For

good performance of the ensemble and strong theoretical guarantees, the individual decision trees must be relatively uncorrelated from one another. Breiman's random forest algorithm does this in two ways:

1. At every node in the decision tree, the optimal split is determined over a random subset $d$ of the total collection of features $p$.

2. Each tree is trained on a randomly bootstrapped sample of data points $D' \subset D$ from the full training data set.

Applying these techniques means that random forests do not overfit and lowers the upper bound of the generalization error (Breiman, 2001).

## 2.3 SPARSE PROJECTION OBLIQUE RANDOMER FORESTS

SPORF is a recent modification to random forest that has shown improvement over other versions (Tomita et al., 2015; Tomita et al., 2017). Recall that RF split nodes partition data along the coordinate axes by comparing the projection $e_j^\mathsf{T} x_i$ of observation $x_i$ on standard basis $e_j$ to a threshold value $\tau$. SPORF generalizes the set of possible projections, allowing for the data to be partitioned along axes specified by any sparse vector $a$.

$$S_\theta^L = \{(x_i, y_i) \mid a^\mathsf{T} x_i < \tau\}$$
$$S_\theta^R = \{(x_i, y_i) \mid a^\mathsf{T} x_i \geq \tau\}$$

Rather than partitioning the data solely along the coordinate axes (i.e. the standard basis), SPORF creates partitions along axes specified by sparse vectors. In other words, let the dictionary $\mathcal{A}$ be the set of atoms $\{a\}$, each atom a $p$-dimensional vector defining a possible projection $a^\mathsf{T} x_i$. In axis-aligned forests, $\mathcal{A}$ is the set of standard basis vectors $\{e_j\}$. In SPORF, the dictionary $\mathcal{D}$ can be much larger, because it includes, for example, all 2-sparse vectors. At each split node, SPORF samples $d$ atoms from $\mathcal{D}$ according to a specified distribution. By default, each of the $d$ atoms are randomly generated with a sparsity level drawn from a Poisson distribution with a specified rate $\lambda$. Then, each of the non-zero elements are uniformly randomly assigned either $+1$ or $-1$. Note that the size of the dictionary for SPORF is $3^p$ (because each of the $p$ elements could be $-1$, $0$, or $+1$), although the atoms are sampled from a distribution heavily skewed towards sparsity.

## 3 METHODS

### 3.1 RANDOM PROJECTION FORESTS ON MANIFOLDS

In the structured setting, the dictionary of projection vectors $\mathcal{A} = \{a\}$ is modified to take advantage of the underlying manifold on which the data lies. We term this method the Manifold Forest (MORF).

Each atom $a$ projects an observation to a real number and is designed with respect to prior knowledge of the data manifold. Nonzero elements of $a$ effectively select and weight features. Since the feature space is structured, each element of $a$ maps to a location on the underlying manifold. Thus, patterns of contiguous points on the manifold define the atoms of $\mathcal{A}$; the distribution of those patterns yields a distribution over the atoms. At each node in the decision tree, MORF samples $d$ atoms, yielding $d$ new features per observation. MORF proceeds just like SPORF by optimizing the best split according to the Gini index. Algorithm pseudocode, essentially equivalent to that of SPORF, can be found in the Appendix.

In the case of two-dimensional arrays, such as images, an observation $x_i \in \mathbb{R}^p$ is a vectorized representation of a data-matrix $X_i \in \mathbb{R}^{W \times H}$. To capture the relevance of neighboring pixels, MORF creates projections by summing the intensities of pixels in rectangular patches. Thus the atoms of $\mathcal{A}$ are the vectorized representations of these rectangular patches.

A rectangular patch is fully parameterized by the location of its upper-left corner $(u, v)$, its height $h$, and width $w$. To generate a patch, first the index of the upper left corner is uniformly sampled. Then its height and width are independently sampled from separate uniform distributions. MORF hyperparameters determine the minimum and maximum heights heights $\{h_{min}, h_{max}\}$ and widths $\{w_{min}, w_{max}\}$, respectively, to sample from. Let $unif\{\alpha, \beta\}$ denote the discrete uniform distribution. An atom $a$ is sampled as follows. Note that the patch cannot exceed the

data-matrix boundaries.

$$u \sim unif\{0, W - w_{min}\} \qquad\qquad v \sim unif\{0, H - h_{min}\}$$
$$w \sim unif\{w_{min}, min(w_{max}, W - u)\} \qquad\qquad h \sim unif\{h_{min}, min(h_{max}, H - v)\}$$

The vectorized atom $a$ yields a projection of the data $a^\mathsf{T} x_i$, effectively selecting and summing pixel intensities in the sampled rectangular patch.

$$a^\mathsf{T} = (\underbrace{0, \ldots, 0}_{W \times v}, \underbrace{0, \ldots, 0}_{u}, \overbrace{\underbrace{1, \ldots, 1}_{w}, \underbrace{0, \ldots, 0}_{W-w}, \ldots, \underbrace{1, \ldots, 1}_{w}, \underbrace{0, \ldots, 0}_{W-w}}^{(h-1) \times W}, \underbrace{1, \ldots, 1}_{w}, \underbrace{0, \ldots, 0}_{W-w-u}, \underbrace{0, \ldots, 0}_{W \times (H-h-v)})$$

By constructing features in this way, MORF learns low-level features in the structured data, such as edges or corners in images. The forest can therefore learn the features that best distinguish a class. The structure of these atoms is flexible and task dependent. In the case of data lying on a cyclic manifold, the atoms can wrap-around borders to capture the added continuity. Atoms can also be used in one-dimensional arrays, such as univariate time-series data, in which case $h_{min} = h_{max} = 1$

## 3.2 FEATURE IMPORTANCE

One of the benefits to decision trees is that their results are fairly interpretable in that they allow for estimation of the relative importance of each feature. Many approaches have been suggested (Breiman, 2001; Lundberg & Lee, 2017) and here a projection forest specific metric is used in which the number of times a given feature was used in projections across the ensemble of decision trees is counted. A decision tree $T$ is composed of many nodes $k$, each one associated with an atom $a_k^*$ and threshold that partition the feature space according to the projection $a_k^* \cdot x_i$. Thus, the indices corresponding to nonzero elements of $a_k^*$ indicate important features used in the projection. For each feature $j$, the number of times $\pi_j$ it is used in a projection, across all split nodes and decision trees, is counted.

$$\pi_j = \sum_T \sum_{k \in T} \mathbb{I}(a_{kj}^* \neq 0)$$

These normalized counts represent the relative importance of each feature in making a correct classification. Such a method applies to both SPORF and MORF, although different results between them would be expected due to different projection distributions yielding different hyperplanes.

## 4 SIMULATION RESULTS

To test MORF, we evaluate its performance in three simulation settings as compared to logistic regression (Log. Reg), linear support vector machine (Lin. SVM), support vector machine with a radial basis function kernel (SVM), k-nearest neighbors (kNN), random forest (RF), Multi-layer Perceptron (MLP), and SPORF (SPORF). For each experiment, we used our open source implementation of MORF and that of SPORF. All decision forest algorithms used 100 decision trees on the simulations. Each of the other classifiers were run from the Scikit-learn Python package (Pedregosa et al., 2011) with default parameters. Additionally, we tested against a Convolutional Neural Network (CNN) built using PyTorch (Paszke et al., 2017) with two convolution layers, ReLU activations, and maxpooling, followed by dropout and two densely connected layers. The CNN results were averaged over 5 runs for the simulations and training was stopped early if the loss plateaued.

## 4.1 SIMULATION SETTINGS

Experiment (A) is a non-Euclidean example inspired by Younes (2018). Each observation is a discretization of a circle into 100 features with two non-adjacent segments of 1's in two differing patterns: class 1 features two segments of length five while class 2 features one segment of length four and one of length six. MORF chose one-dimensional rectangles in this setting as the observations were one-dimensional in nature. These projection patches had a width between one and fifteen pixels and each split node of SPORF and MORF considered 40 random projections. Figure 1(A) shows examples from the two classes and classification results across various sample sizes.

In experiment (B) consists of a simple $28 \times 28$ binary image classification problem. Images in class 0 contain randomly sized and spaced *horizontal* bars while those in class 1 contain randomly sized and spaced *vertical* bars. For each sampled image, $k \sim Poisson(\lambda = 10)$ bars were distributed among the rows or columns, depending on the class. The distributions of the two classes are identical if a 90 degree rotation is applied to one of the classes. Projection patches were between one and four pixels in both width and height and each split node of SPORF and MORF considered 28 random projections. Figure 1(B) shows examples from the two classes and classification results across various sample sizes.

Experiment (C) is a signal classification problem. One class consists of 100 values of Gaussian noise while the second class has an added exponentially decaying unit step beginning at time 20.

$$x_t^{(0)} = \epsilon_t$$
$$x_t^{(1)} = u(t - 20)e^{(t-20)} + \epsilon_t$$
$$\epsilon_t \sim \mathcal{N}(0, 1)$$

Projection patches were 1D with a width between one and five timesteps. Each split node of SPORF and MORF considered the default number of random projections, the square root of the number of features. Figure 1(C) shows examples from the two classes and classification results across various sample sizes.

## 4.2 CLASSIFICATION ACCURACY

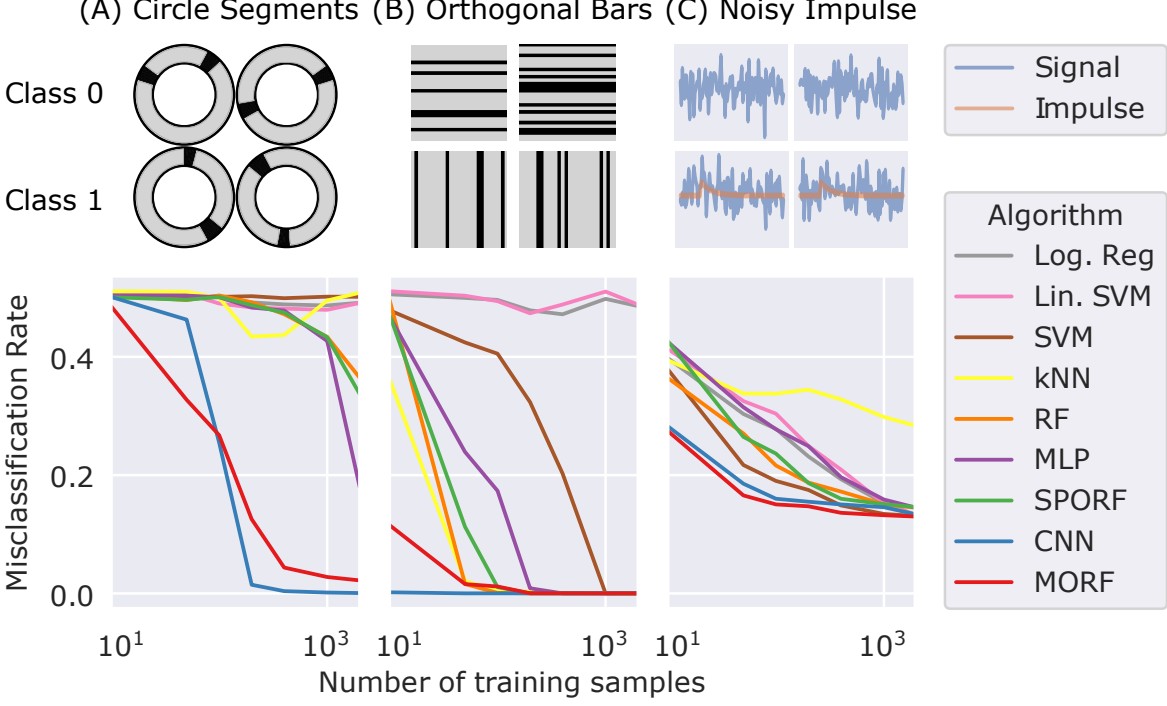

Figure 1: MORF outperforms other algorithms in three two-class classification settings. Upper row shows examples of simulated data from each setting and class. Lower row shows misclassification rate in each setting, tested on 10,000 test samples. **(A)** Two segments in a discretized circle. Segment lengths vary by class. **(B)** Image setting with uniformly distributed horizontal or vertical bars. **(C)** White noise (class 0) vs. exponentially decaying unit impulse plus white noise (class 1).

In all three simulation settings, MORF outperforms all other classifiers, doing especially better at low sample sizes, except the CNN for which there is no clear winner. The performance of MORF is particularly good in the discretized circle simulation for which most other classifiers perform at chance levels. MORF also performs well in the signal

classification problem although all the classifiers are close in performance. This may be because the exponential signal is prevalent throughout most of the time-steps and so perfect continuity is less relevant.

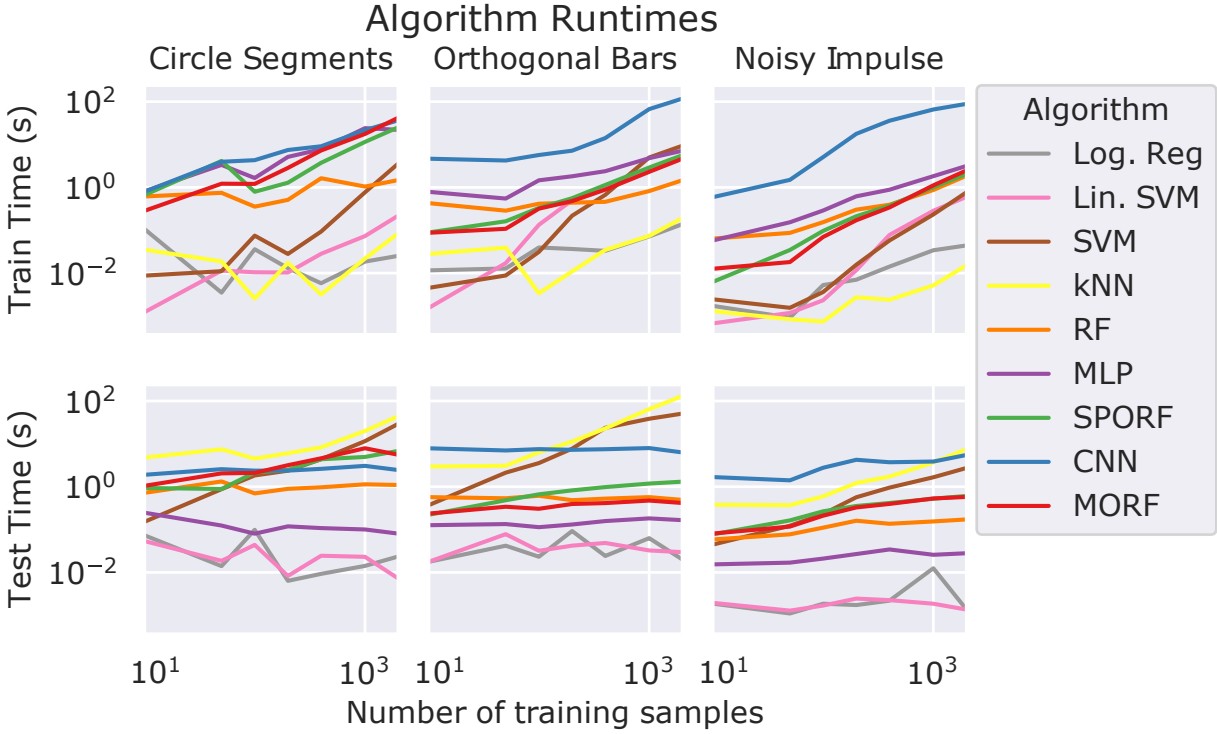

Figure 2: Algorithm train times (above) and test times (below). MORF runtime is not particularly costly and well below CNN runtime in most examples.

### 4.3 RUN TIME

All experiments were run on a single core CPU. MORF has train and test times on par with those of SPORF and RF and so is not particularly more computationally intensive to run. The CNN, however, took noticeably longer to run—especially in terms of training, but also in testing—in two of the three simulations. Thus its strong performance in those settings comes at an added computational cost, a typical issue for deep learning methods (Livni et al., 2014).

## 5 REAL DATA RESULTS

### 5.1 CLASSIFICATION ACCURACY

MORF's performance was evaluated on the MNIST dataset, a collection of handwritten digits stored in 28 by 28 square images (Lecun et al.), and compared to the algorithms used in the simulations. 10,000 images were held out for testing and a subset of the remaining images were used for training. The results are displayed in Figure 3. All three forest algorithms were composed of 500 decision trees and MORF was restricted to use patches up to only 3 pixels in height and width. MORF showed an improvement over the other algorithms, especially for smaller sample sizes. Thus, even this trivial modification can improve performance by several percentage points. Specifically, MORF achieved a better (lower) classification error than all other algorithms besides CNNs for all sample sizes on this real data problem.

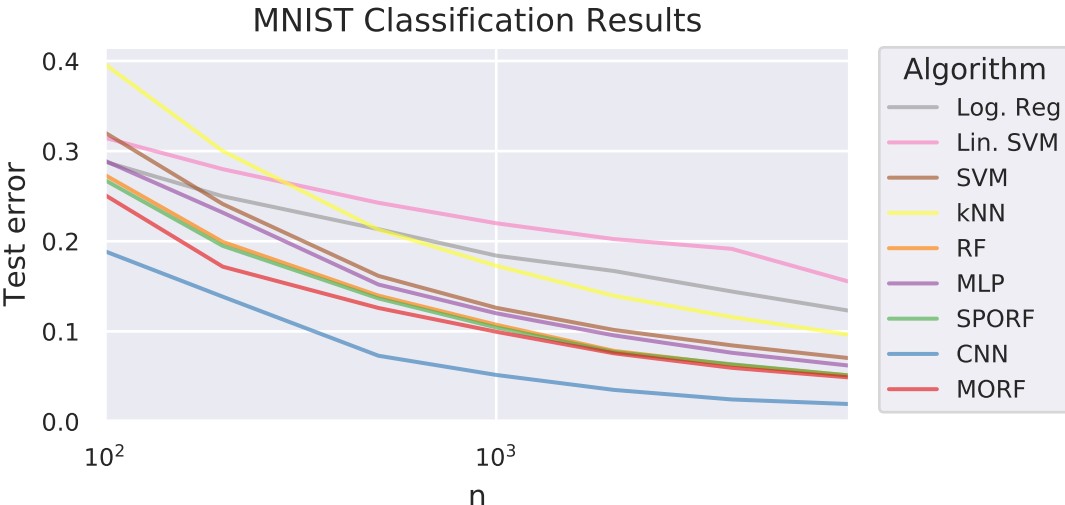

Figure 3: MORF improves classification accuracy over all other non-CNN algorithms for all sample sizes, especially in small sample sizes.

## 5.2 FEATURE IMPORTANCE

To evaluate the capability of MORF to identify importance features in manifold-valued data as compared to SPORF and RF. All methods were run on a subset of the MNIST dataset: we only used threes and fives, 100 images from each class.

The feature importance of each pixel is shown in Figure 4. MORF visibly results in a smoother pixel importance, a result most likely from the continuity of neighboring pixels in selected projections. Although Tomita et al. (2015) demonstrated empirical improvement of SPORF over RF on the MNIST data, its projection distribution yields scattered importance of unimportant background pixels as compared to RF. Since projections in SPORF have no continuity constraint, those that select high importance pixels will also select pixels of low importance by chance. This may be a nonissue asymptotically, but is a relevant problem in low sample size settings. MORF, however, shows little or no importance of these background pixels by virtue of the modified projection distribution.

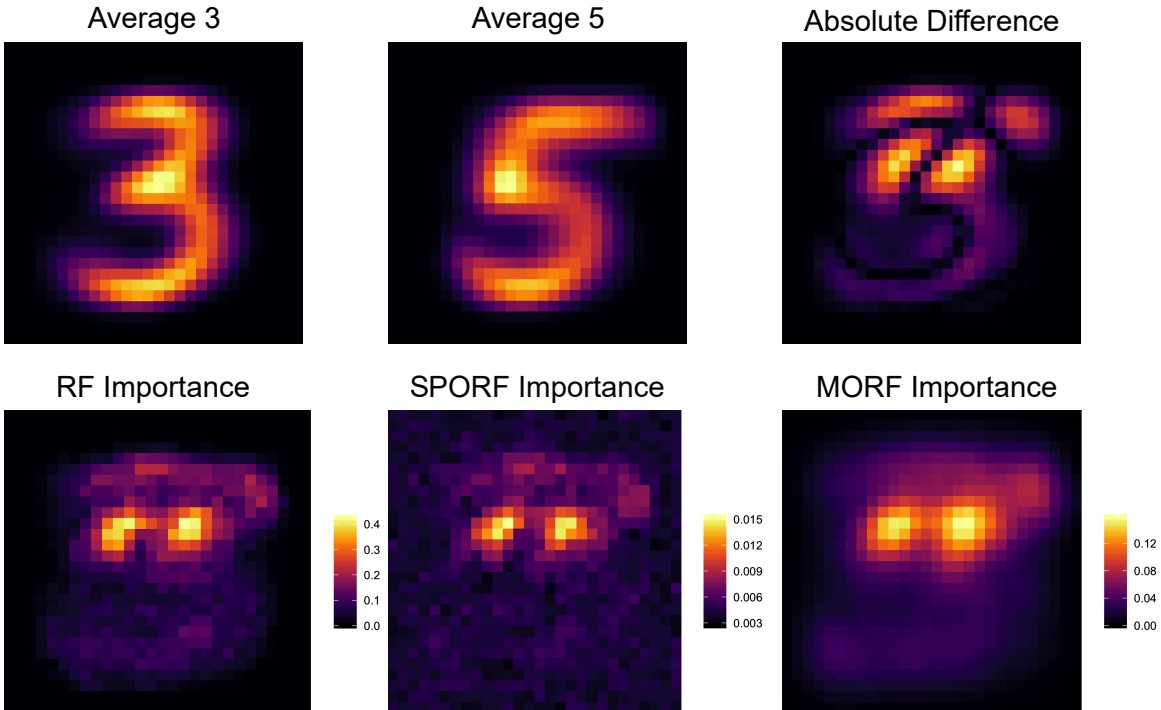

Figure 4: Averages of images in the two classes and their difference (above). Feature importance from MORF (bottom right) shows less noise than SPORF (bottom middle) and is smoother than RF (bottom left).

## 6 DISCUSSION

The success of sparse oblique projections in decision forests has opened up many possible ways to improve axis-aligned decision forests (including random forests and gradient boosting trees) by way of specialized projection distributions. Traditional decision forests have already been applied to structured data, using predefined features to classify images or pixels. Decision forest algorithms like that implemented in the Microsoft Kinect showed great success but ignore pixel continuity and specialize for a specific data modality, namely images (Shotton et al., 2011).

We expanded upon sparse oblique projections and introduced a structural projection distribution that uses prior knowledge of the topology of a feature space. The open source implementation of SPORF has allowed for a relatively easy implementation of MORF, creating a flexible classification method for a variety of data modalities. We showed in various simulated settings that appropriate domain knowledge can improve the projection distribution to yield impressive results that challenge the strength of deep learning techniques on manifold-valued data. On the MNIST data set MORF showed modest improvements over the other algorithms besides CNNs and smoother importance plots than the other decision forest algorithms. This is in spite of the the data set's low resolution images which are harder for the modified projection distributions to take advantage of.

Research into other, task-specific convolution kernels may lead to improved results in real-world computer vision tasks. Such structured projection distributions, while incorporated into SPORF here, may also be incorporated into other state of the art algorithms such as XGBOOST (Chen & Guestrin, 2016).

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

# 7 APPENDIX

---

**Algorithm 1** Learning a Manifold Forest decision tree.

---

**Input:** (1) $\mathcal{D}_n$: training data (2) $d$: dimensionality of the projected space, (3) $f_{\mathbf{A}}$: distribution of the atoms, (4) $\Theta$: set of split eligibility criteria

**Output:** A MORF decision tree $T$

 1: **function** $T = \text{GROWTREE}(\mathbf{X}, \mathbf{y}, f_{\mathbf{A}}, \Theta)$
 2:    $c = 1$                                                              $\triangleright$ $c$ is the current node index
 3:    $M = 1$                                  $\triangleright$ $M$ is the number of nodes currently existing
 4:    $S^{(c)} = \text{bootstrap}(\{1, ..., n\})$            $\triangleright$ $S^{(c)}$ is the indices of the observations at node $c$
 5:    **while** $c < M + 1$ **do**                          $\triangleright$ visit each of the existing nodes
 6:        $(\mathbf{X}', \mathbf{y}') = (\mathbf{x}_i, y_i)_{i \in S^{(c)}}$               $\triangleright$ data at the current node
 7:        **for** $k = 1, \ldots, K$ **do** $n_k^{(c)} = \sum_{i \in S^{(c)}} I[y_i = k]$ **end for**      $\triangleright$ class counts (for classification)
 8:        **if** $\Theta$ satisfied **then**                          $\triangleright$ do we split this node?
 9:            $\mathbf{A} = [\mathbf{a}_1 \cdots \mathbf{a}_d] \sim f_{\mathbf{A}}$           $\triangleright$ sample random $p \times d$ matrix of atoms
10:           $\widetilde{\mathbf{X}} = \mathbf{A}^T \mathbf{X}' = (\widetilde{\mathbf{x}}_i)_{i \in S^{(c)}}$       $\triangleright$ random projection into new feature space
11:           $(j^*, t^*) = \text{findbestsplit}(\widetilde{\mathbf{X}}, \mathbf{y}')$                    $\triangleright$ Algorithm 2
12:           $S^{(M+1)} = \{i : \mathbf{a}_{j^*} \cdot \widetilde{\mathbf{x}}_i \leq t^* \quad \forall i \in S^{(c)}\}$       $\triangleright$ assign to left child node
13:           $S^{(M+2)} = \{i : \mathbf{a}_{j^*} \cdot \widetilde{\mathbf{x}}_i > t^* \quad \forall i \in S^{(c)}\}$      $\triangleright$ assign to right child node
14:           $\mathbf{a}^{*(c)} = \mathbf{a}_{j^*}$                         $\triangleright$ store best projection for current node
15:           $\tau^{*(c)} = t^*$                       $\triangleright$ store best split threshold for current node
16:           $\kappa^{(c)} = \{M + 1, M + 2\}$           $\triangleright$ node indices of children of current node
17:           $M = M + 2$                    $\triangleright$ update the number of nodes that exist
18:        **else**
19:           $(\mathbf{a}^{*(c)}, \tau^{*(c)}, \kappa^{*(c)}) = \text{NULL}$
20:        **end if**
21:        $c = c + 1$                                     $\triangleright$ move to next node
22:    **end while**
23:    **return** $(S^{(1)}, \{\mathbf{a}^{*(c)}, \tau^{*(c)}, \kappa^{(c)}, \{n_k^{(c)}\}_{k \in \mathcal{Y}}\}_{c=1}^{m-1})$
24: **end function**

---

**Algorithm 2** Finding the best node split. This function is called by growtree (Alg 1) at every split node. For each of the $p$ dimensions in $\mathbf{X} \in \mathbb{R}^{p \times n}$, a binary split is assessed at each location between adjacent observations. The dimension $j^*$ and split value $\tau^*$ in $j^*$ that best split the data are selected. The notion of "best" means maximizing some choice in scoring function. In classification, the scoring function is typically the reduction in Gini impurity or entropy. The increment function called within this function updates the counts in the left and right partitions as the split is incrementally moved to the right.

---

**Input:** (1) $(\mathbf{X}, \mathbf{y}) \in \mathbb{R}^{p \times n} \times \mathcal{Y}^n$, where $\mathcal{Y} = \{1, \ldots, K\}$
**Output:** (1) dimension $j^*$, (2) split value $\tau^*$

1: **function** $(j^*, \tau^*) = \text{FINDBESTSPLIT}(\mathbf{X}, \mathbf{y})$
2:     **for** $j = 1, \ldots, p$ **do**
3:         Let $\mathbf{x}^{(j)} = (x_1^{(j)}, \ldots, x_n^{(j)})$ be the $jth$ row of $\mathbf{X}$.
4:         $\{m_i^j\}_{i \in [n]} = \text{sort}(\mathbf{x}^{(j)})$                                                       ▷ $m_i^j$ is the index of the $i^{th}$ smallest value in $\mathbf{x}^{(j)}$
5:         $t = 0$                                                                                                  ▷ initialize split to the left of all observations
6:         $n' = 0$                                                                                               ▷ number of observations left of the current split
7:         $n'' = n$                                                                                            ▷ number of observations right of the current split
8:         **if** (task is classification) **then**
9:             **for** $k = 1, \ldots, K$ **do**
10:                 $n_k = \sum_{i=1}^{n} I[y_i = k]$                                                     ▷ total number of observations in class $k$
11:                 $n_k' = 0$                                                                                ▷ number of observations in class $k$ left of the current split
12:                 $n_k'' = n_k$                                                                             ▷ number of observations in class $k$ right of the current split
13:             **end for**
14:         **end if**
15:         **for** $t = 1, \ldots, n - 1$ **do**                                                  ▷ assess split location, moving right one at a time
16:             $(\{(n_k', n_k'')\}, n', n'', y_{m_t^j}) = \text{increment}(\{(n_k', n_k'')\}, n', n'', y_{m_t^j})$
17:             $Q^{(j,t)} = \text{score}(\{(n_k', n_k'')\}, n', n'')$                               ▷ measure of split quality
18:         **end for**
19:     **end for**
20:     $(j^*, t^*) = \underset{j,t}{\text{argmax}}\, Q^{(j,t)}$
21:     **for** $i = 0, 1$ **do** $c_i = m_{t^*+i}^{j^*}$ **end for**
22:     $\tau^* = \frac{1}{2}(x_{c_0}^{(j^*)} + x_{c_1}^{(j^*)})$                                         ▷ compute the actual split location from the index $j^*$
23:     **return** $(j^*, \tau^*)$
24: **end function**

---

