# OpenReview forum: "MANIFOLD FORESTS: CLOSING THE GAP ON NEURAL NETWORKS"
_ICLR.cc/2020/Conference — Reject_

### Official Review · AnonReviewer2 · 2019-10-22
**Official Blind Review #2**

**Rating:** 3

**Review:**

This paper proposed a new method called manifold forest to improve decision forest (DF) classification results. It is motivated by that, natural data is often in some manifold but not randomly distributed. It showed how to use the 2D spatial structures of natural images by constructing structured atoms. Results on 3 toy examples and MNIST showed the better performance than standard RF and SPORF.

Overall, the paper is easy to follow and well written. The idea is intuitive: using structured 2D information to improve the classification results. But there are some issues with the implemetation.

1. In image classification, we definitely need 2D structure information. This is normally extracted by the descriptors such as SIFT, GIST, where the 2D information has been included. It is rare to use the pixel values as the features directly for classification.   In this case, the benefit of the proposed method is very weak. This is the main issue of the paper. No results on real features.

2. The results are weak too. The real data is MNIST, which is also a very toy dataset. It would be good to include some real world image dataset, such as CIFAR, ImageNet etc.

3. The algorithm is somehow incremental compared with SPORF.


**Experience Assessment:**

I have published in this field for several years.

**Review Assessment: Checking Correctness Of Derivations And Theory:**

I did not assess the derivations or theory.

**Review Assessment: Checking Correctness Of Experiments:**

I assessed the sensibility of the experiments.

**Review Assessment: Thoroughness In Paper Reading:**

I read the paper thoroughly.

---

### Official Review · AnonReviewer1 · 2019-10-26
**Official Blind Review #1**

**Rating:** 1

**Review:**

The goal of the paper is clear. However, the proposed method has only a very incremental novelty compared to SPORF and the previous approaches in computer vision (e.g., Shotton et al., 2011). Although the authors claim the method can take advantage of structure in all kinds of data, the only conducted experiment is on the image data which is fairly limited.

In Fig. 1, they claim MORF outperforms other methods given fewer training data. However, for classifying Orthogonal Bars, CNN still outperforms MORF when few training data are given. The results on MNIST is also not impressive. CNN is still much better compared to other tree-based methods. Moreover, no other real-world dataset has been conducted experiments on.

In general, the paper's goal is clear and interesting. But the authors failed to propose a novel method and the results are not convincing. Hence, I recommend rejection.


**Experience Assessment:**

I have published one or two papers in this area.

**Review Assessment: Checking Correctness Of Derivations And Theory:**

I assessed the sensibility of the derivations and theory.

**Review Assessment: Checking Correctness Of Experiments:**

I assessed the sensibility of the experiments.

**Review Assessment: Thoroughness In Paper Reading:**

I read the paper at least twice and used my best judgement in assessing the paper.

---

### Official Review · AnonReviewer4 · 2019-10-31
**Official Blind Review #4**

**Rating:** 3

**Review:**



====== Post Rebuttal ======

No rebuttal was provided and all reviewers have raised issues. Therefore, I will maintain the original rating.

====== Summary======

The paper puts forward a potential issue with the standard decision tree and tries to remedy it. The issue is that standard decision trees, by independently picking the split feature(s), virtually discard the structure of the input features (if any). In this work, structure essentially refers to ordered input features such as sequences, signals, and images. The work is motivated by the fact that convolutional networks (ConvNet) exploit this structure thanks to its local convolution operation. The idea of this work is to bridge this gap by constraining the split feature selection to follow the local structures of the input where each split feature would correspond to a local bounding box of the input feature. Applying this idea to random forests, it demonstrates significant improvements over unstructured split features, on a few synthesized datasets as well as MNIST dataset. The performance is also compared to one ConvNet architecture which demonstrates similar performance on synthesized datasets while being considerably slower than the proposed random forests.


====== Strengths and Weaknesses ======

+ The motivation to make random forests respect the input structure similar to ConvNets is well-grounded and important since random forests are still being used in certain applications where computational complexity and/or interpretability are crucial factors.
+ It proposes a simple technique to add locality to the split criterion which brings significant improvements over the standard random forest.

- I believe the paper’s title should be closing the gap to *convolutional* neural networks since fully-connected networks are not local in the first place.
- continuing on the previous point, the proposed method pushes the decision forests closer to “locally-connected” networks where, although the features are local, they are not shared across different locations in the input. The additional “sharing” property which takes locally-connected networks to convolutional networks is an important property of ConvNets since it brings translation equivariance for the representations. The proposed random forest method is not translation-equivariant by design and is only local.
- regarding the previous point, a larger difference between ConvNet and the proposed method is more imminent if one goes to datasets with “non-aligned” observations. This already becomes more evident in the MNIST (which contains mostly aligned digits) where ConvNet clearly outperform MORF but would likely become more significant when going to real-world datasets, e.g. CIFAR.

- the proposed method is very similar to patch-based random forest image recognition methods. For instance, several variants exist that are used for object or part detection in a given image where a patch is selected from the image for the split criterion. This patch will respect locality in a similar way to the proposed MORF’s bounding boxes. For instance, see “tomography scans” example of Criminisi et al. 2012 (section 4.5).

- The paper is missing to provide many important details
- what are the actual hyperparameter (hp) values used for the different methods in figure 1,2, and 3. This includes the hp relevant for the random forests including the number of decision trees, the stopping criterion, the random data partitioning method, number of random projections, as well as h_min, h_max, w_min, and w_max. It also does not discuss the hyper parameters of the baseline methods including k for KNN, distance measure for KNN, penalty cost for SVM, variance for the RBF kernel, ConvNet architecture, etc.
- more importantly, it is not mentioned how these hp are optimized for the different baselines as well as the proposed method. What algorithm has been used (e.g. grid search)? How much hp optimization budget is used for different baselines? Is there a validation set put aside for hp optimization?
- from the description in the start of page 4, it seems that the atoms for the proposed MORF are vectors of binary dimensions while for the general SPORF each atom’s element can be -1 as well (page 3). Why is this choice made despite the fact that it reduces the capacity of the model?

- the bounding box sampling seems biased as presented. That is, bounding boxes closer (than h_max and/or w_max) to the right and/or lower borders are more likely since the number of valid boxes will be lower.


====== Final Decision ======

I think it will be very interesting to bridge the gap between ConvNets and random forests since the latter comes with attractive properties. While I find all the concerns that are listed above important, my rating is mainly due to the novelty of this work compared to the prior patch-based random forest techniques for image analysis.


====== Points of improvements ======

I believe it’s important to disentangle the two main properties of convolution operation in ConvNets being shared and local parameters. Then, accordingly propose strategies to bring these properties to a decision tree.



**Experience Assessment:**

I have published one or two papers in this area.

**Review Assessment: Checking Correctness Of Derivations And Theory:**

I carefully checked the derivations and theory.

**Review Assessment: Checking Correctness Of Experiments:**

I carefully checked the experiments.

**Review Assessment: Thoroughness In Paper Reading:**

I read the paper thoroughly.

---

### Decision · Program_Chairs · 2019-12-19

**Decision:**

Reject

**Comment:**

This work explores how to leverage structure of this input in decision trees, the way this is done for example in convolutional networks.
All reviewers agree that the experimental validation of the method as presented is extremely weak. Authors have not provided a response to answer the many concerns raised by reviewers.
Therefore, we recommend rejection.